# Chemical Composition and Biological Activity of Five Essential Oils from the Ecuadorian Amazon Rain Forest

**DOI:** 10.3390/molecules24081637

**Published:** 2019-04-25

**Authors:** Paco Noriega, Alessandra Guerrini, Gianni Sacchetti, Alessandro Grandini, Edwin Ankuash, Stefano Manfredini

**Affiliations:** 1Group of Research and Development in Sciences Applied to Biological Resources, Universidad Politécnica Salesiana, Avenida 12 de Octubre N 2422 y Wilson, Quito 170109, Ecuador; 2Kutukú Biological Station, Universidad Politécnica Salesiana, Sevilla Don Bosco Parish, Macas 140150, Ecuador; 3Department of Life Sciences and Biotechnology, University of Ferrara, Pharmaceutical Biology Lab., Technopole Lab. Terra&Acqua Tech (Research Unit 7), P.le Luciano Chiappini 3, Malborghetto di Boara, 44123 Ferrara, Italy; scg@unife.it (G.S.); alessandro.grandini@unife.it (A.G.); 4Shakaim Biological Station, Chiguaza Parish 140751, Ecuador; expediciones.shakaim@hotmail.com; 5Department of Life Sciences and Biotechnology, Master Course in Cosmetic Science and Technology (COSMAST), University of Ferrara, Via L. Borsari 46, 44121 Ferrara, Italy; mv9@unife.it

**Keywords:** Amazonian essential oils, biological activity, GC-MS, antimicrobial activity, antioxidant activity

## Abstract

The chemical composition and biological activity of essential oils isolated from the leaves of *Siparuna aspera*, *Siparuna macrotepala*, *Piper leticianum*, *Piper augustum* and the rhizome of *Hedychium coronarium* were evaluated. These species are used medicinally in different ways by the Amazonian communities that live near the Kutukú mountain range. Chemical studies revealed that the main components for the two *Siparuna* species were germacrene D, bicyclogermacrene, α-pinene, δ-cadinene, δ-elemene, α-copaene and β-caryophyllene; for the two *Piper* species β-caryophyllene, germacrene D, α-(*E,E*)-farnesene, β-elemene, bicyclogermacrene, δ-cadinene and for *H. coronarium* 1,8-cineole, β-pinene, α-pinene and α-terpineol. The antioxidant activity of all essential oils was evaluated by 1,1-diphenyl-2-picrylhydrazyl (DPPH), 2,2′-azino-bis(3-ethylbenzothiazoline-6-sulfonic acid) diammonium salt (ABTS), photochemiluminescence (PCL) quantitative assays, and DPPH and ABTS bioautographic profiles, with different results for each of them. Antimicrobial activity studies were carried out on three yeasts, six Gram positive and four Gram negative bacteria, by means of the disc diffusion method. The essential oil of *H. coronarium* showed the most relevant results on *L. grayi*, *K. oxytoca* and *S. mutans*, *P. augustum* and *P. leticianum* on *S. mutans*. An antibacterial bioautographic test for *H. coronarium* was also carried out and highlighted the potential activity of terpinen-4-ol and 1,8-cineole.

## 1. Introduction

According to the Monitoring Center for the Conservation of the Environment, Ecuador, with just 0.19% of the surface of the Earth, is one of 17 mega-diverse countries, home to 10% of all plant species in the world [1]. Additionally, in Ecuador, diverse ancestral peoples are possessors of millenary knowledge about the use and management of these resources. The Shuar, with some 110,000 people, are one of these populations, whose members reside mainly in the provinces of Morona Santiago, Pastaza and Zamora Chinchipe [2]. All these arguments make it necessary to corroborate and to valorize the uses of medicinal plants within the Shuars’ region. For the present investigation, five species, used as medicines by the Shuar indigenous people living near the Kutukú mountain range, in the Morona Santiago province, were selected.

*Siparuna aspera* (Ruiz & Pav.) A. DC., is a plant native to Ecuador, which can also be found in Bolivia, Colombia, Peru and Venezuela [3]. It is commonly known as “limoncillo” and among the Shuar it is called “mejentsuna”. The infusion of its leaves is used for lowering fevers. *Siparuna macrotepala* Perkins, is a plant native to Ecuador that is also distributed in Colombia and Peru [3], whose common name is “limoncillo”, known among the Shuar as “tTsuna”-. The infusion of the leaves is used to fight malaria and influenza. *Piper leticianum* C. DC., a plant native to Ecuador, also found in Colombia and Peru [4], is known as “untuntuntup” among the Shuar, that use the leaves to fight tooth decay. *Piper augustum* Rudge, a plant native to Ecuador, is found throughout tropical America [5]; the Shuar ethnic group calls it “untuntup” and it is used to fight tooth decay. *Hedychium coronarium* J. Koenig, a plant native to tropical Asia, is an introduced species in Ecuador with the common name of “lirio de muerto” and known among the Shuar as “ajejà”. It is used for its analgesic, antiseptic and digestive properties.

The scientific literature reports few chemical and biological studies carried out on these native species. In the case of *S. aspera*, there is a work that evaluates the leishmanicidal properties of the plant extracts [6]. For *S. macrotepala* a chemical study denotes the presence of cadinane-sesquiterpenes [7]. Regarding *P. augustum,* an investigation of the chemical composition of the essential oil, isolated from its leaves, reports α-phellandrene (14.7%), β-caryophyllene (13.5%), limonene (13.0%), α-pinene (10.5%) and linalool (10.3%) as main components [8]. A second study assessed its anti-leishmanial activity [9]. For *P. leticianum,* no investigations have been found.

Of the five species studied, *H. coronarium* is the only species about which the scientific literature presents more information, highlighting its essential oil isolated from various parts of the world such as China [10], in the Himalayas [11], Brazil [12] and India [13]. Comparing the chemical studies, it is possible to appreciate a diversity in its components depending on the origin of the species. From a pharmacological point of view, there are studies that confirm its analgesic [14], antibacterial [11,13,15], anti-inflammatory [16], anthelmintic [17], antioxidant [18,19], cytotoxic [20] and hepatoprotective [21] activity.

One of the objectives of the present investigation is to reach a chemical characterization of the essential oils of the five Amazonian species; especially for *S. aspera* and *P. leticianum*, whose chemical composition was never previously described. A second objective is related to the evaluation of in vitro biological activity, performing antimicrobial and antioxidant activity studies. The research described in this manuscript ultimately is driven by the need to valorize the biodiversity and the knowledge of the native peoples of Ecuador, for their own benefits based on sustainable exploitation and in general for the advantage of the Ecuadorian population. Some of these medicinal plants could be used as an alternative health products to synthetic medicines, whose economic benefits generally fall to transnational pharmaceutical corporations.

## 2. Results

### 2.1. Essential Oils Yield and Density

The essential oils yield (%*w*/*w*) and density were as follows: *S. aspera*, a yield of 0.15% and a density of 0.929 g /mL; *S. macrotepala*, a yield of 0.15% and a density of 0.930 g/mL; *P. augustum* a yield of 0.02% and a density of 0.908 g /mL; *P. leticianum* a yield of 0.02% and a density of 0.905 g/mL; *H*. *coronarium* a yield of 0.04% and a density of 0.895 g/mL.

### 2.2. Chemical Composition

The detailed chemical compositions of each essential oil, determined above 90%, can be seen in Table 1.

#### 2.2.1. DPPH and ABTS Assays

The IC_50_ values are expressed in mg/mL and represent the concentration capable of inhibiting 50% of the oxidation of DPPH and ABTS. The *Thymus vulgaris* essential oil was used as a positive control, the activity of some pure compounds was also evaluated. The IC_50_ of germacrene D, separated by silica gel column, was performed: the results are shown in Table 2.

#### 2.2.2. PCL Photochemiluminescence

Due to the lipophilic nature of the essential oils the method used is the ACL, in which the activity is expressed as μmol of Trolox/mL (Table 3), and the positive control is *T. vulgaris* essential oil.

#### 2.2.3. HPTLC Antiradical Bioautographic Assay with DPPH and ABTS

The results revealed a high antioxidant activity of the *Siparuna* and *Piper* species, corresponding to the fraction with Rf = 0.8 where we found germacrene D and β-caryophyllene, for *H. coronarium* the activity was observed in the band Rf = 0 corresponding to 1,8-cineole (Figure 1).

### 2.3. Evaluation of the Minimum Inhibitory Concentration (MIC) 

The results of the minimum inhibitory concentration are expressed in mg/mL and shown in Table 4.

#### Bioautographic Antibacterial Activity of the Essential Oil of *H. coronarium*

The interesting activity of the essential oil on several bacteria made it necessary to find the components responsible for this effect. In Figure 2 it is showed that 1,8 cineole, terpinen-4-ol explain the activity on the Gram positive bacteria. Another unidentified minor compound seems to contribute to the antibacterial effect.

## 3. Discussion

The chemical composition of *S. aspera* and *P. leticianum* essential oils was studied for the first time in this research. The main molecules in *S. aspera* essential oil were germacrene D (23.2%), bicyclogermacrene (7.8%) and α-pinene (7.0%). In a previously study we found germacrene D and bicyclogermacrene as major compounds in *Siparuna schimpffi* spice [24] and α-pinene as a minor component. For *S. macrotepala*, the presence of cadinane compounds in interesting amounts was evidenced [7], not revealed in our essential oil. *P. leticianum* essential oil showed β-caryophyllene (21.8%) and germacrene D (9.0%) as main compounds, and *P. augustum* essential oil had a similar composition. Studies of other Piperaceae family species, like *P. nigrum* [25], *P. marginatum* [26] and *P. cernuum* [27], highlighted a high β-caryophyllene content. Research on *P. augustum* essential oil confirmed β-caryophyllene as the main molecule [8,28]. Numerous studies in the essential oil from *H. coronarium* rhizomes reported very similar chemical compositions, with 1,8-cineole and β-pinene as main components [12,13], with exception of the data reported by Prakash et al. where linalool and limonene were the principal molecules [11]. The antioxidant activities of the essential oils were not comparable with the positive control, *Thymus vulgaris* essential oil. However, it is important to note the interesting radical scavenging effect of germacrene D, isolated from *S. macrotepala* essential oil, with a DPPH IC_50_ of 2.1 mg/mL and ABTS IC_50_ of 1.1 mg/mL, also shown through the HPTLC bioautographic test. Photochemiluminescence assays confirmed the low antioxidant activity of the essential oils compared to the positive control. Finally, we studied the antimicrobial activity against a wide spectrum of bacteria and yeasts. The essential oil of *H. coronarium* showed an interesting antimicrobial effect vs. *L. grayi*, *K. oxytoca* and *S. mutans* and the literature has documented good results for an essential oil with similar composition against *Trichoderma* sp. and *Candida albicans* [13]. Other paper studied the antimicrobial activity, but the chemical composition was different from our essential oil [11] or the chemical characterization was not investigated [15]. *P. augustum* and *P. leticianum* highlighted interesting results against *S. mutans.* The antimicrobial bioautographic assay of *H. coronarium* essential oil on *S. aureus*, chosen as model for this experimental approach, showed that 1,8-cineole and terpinen-4-ol were the molecules responsible for the activity. Some studies have highlighted the appreciable antimicrobial capacity of these molecules [29,30]. 

## 4. Materials and Methods

### 4.1. Plant Material

Plants were collected in different sites in the province of Morona Santiago, and the coordinates and sectors are listed in Table 5.

For the isolation of essential oils, the method known as hydrodistillation [31], was utilized using 250 L equipment (The Essential Oil Company, Portland, OR, USA) belonging to the Chankuap Resources for the Future Foundation.

### 4.2. GC-MS and GC-FID Analyses

The essential oil composition was determined by gas-chromatography coupled to mass-spectrometry, and the quantification of individual components was performed by GC-FID, calculating the relative peak average area of three separated injections. A Varian 3800 gas chromatograph (Varian, Palo Alto, CA, USA) was used, equipped with a Factor four VF-5ms column (poly-5% phenyl-95% dimethylsiloxane) of 30 m length, with an internal diameter of 0.25 mm and a film of 0.25 μm, directly coupled to a Varian 4000 mass spectrometer. The carrier gas was helium with a flow of 1 mL/min and a split ratio of 1:50. The analysis starts at 45 °C and reaches 100 °C at a rate of 1 °C per minute, then rises to a temperature of 250 °C at a speed of 5 °C, staying at that temperature for 15 min: the total analysis time was 90 min. The conditions of the mass spectrometer were: ionization energy: 70 eV; emission current: 10 μAmp, scan rate: 1 scan/s, mass range: 35–400 Da, trap temperature: 220 °C, transfer line temperature: 260 °C. The identification of compounds were performed by comparing their arithmetic indices (AI) and the MS fragmentation pattern with those of other known essential oils, with pure compounds and by matching the MS fragmentations patterns and arithmetic indices with mass spectra libraries and with those in the literature [22,32]. The experimental arithmetic index of each component was determined adding a C_8_-C_32_ n-alkanes mixture (Sigma-Aldrich Italy, Milano, Italy) to the essential oil before injection in the GC-MS equipment and analyzing it under the same conditions reported above [22]. For the quantitative analysis a ThermoQuest GC-Trace gas-chromatograph (ThermoQuest Italia, Rodano, Italy) equipped with a FID detector and the same column above described were used. The operating conditions for gas chromatograph were reported above. FID temperature was 250 °C. The oil percentage composition was performed by the normalization method from the GC peak areas, without using correction factors [32] and was the average of three injections.

### 4.3. Antioxidant Activity

The antioxidant properties of the 5 essential oils were analyzed by various tests used in studies with essential oils: 1,1-diphenyl-2-picrylhydrazyl (DPPH) and 2,2′-azino-bis(3-ethylbenzo-thiazoline-6-sulfonic acid) diammonium salt (ABTS) quantitative assays [32], DPPH and ABTS HPTLC bioautographic methods [33,34,35], and photochemiluminescence (PCL) tests [36,37]. Data reported for each assay are the average of three independent experiments.

#### 4.3.1. Quantitative Free Radical Scavenging Activity: DPPH and ABTS Assays

For the DPPH test, the essential oils and some of their constituents, germacrene D (isolated from *S. macrotepala*, through silica gel column chromatography using *n*-hexane as mobile phase), β-pinene, 1,8-cineole, *E*-β-caryophyllene were diluted 2, 5, 10, 50, 100, 200-fold in dimethylsulfoxide (DMSO) and an aliquot of 100 μL of each solution (or DMSO, for the blank) was added to 2.9 mL of DPPH (1 × 10^−4^ in ethanol). All solutions were stirred vigorously for 30 min in the dark at room temperature. The absorbances were measured at 517 nm in a Helios spectrophotometer, (Thermo Spectronic, Cambridge, UK). ABTS radical was prepared mixing 10 mL of 2 mM ABTS aqueous solution with 100 μL of 70 mM K_2_S_2_O_8_ aqueous solution: the reaction is complete after 12–16 h, in the dark and at room temperature. 1 mL of the last solution was diluted with ethanol since to achieve an absorbance of 0.70 ± 0.02 at 734 nm. Similarly to DPPH test, we proceeded mixing 10 μL of each diluted essential oils (or DMSO for the blank) with 0.990 mL of ABTS solution. Absorbances were measured at 734 nm. The antiradical activity for each mixture was calculated according to the following formula:Ip DPPH or ABTS% = (A*b* − A*a*)/A*b* × 100(1)where A*b* and A*a* are the absorbances of the blank and the samples respectively after 30 min for DPPH and 1-min for ABTS assay. The antiradical activity of the essential oil is evaluated by calculation of the IC_50_ value, which is equivalent to the concentration providing 50% of the DPPH or ABTS inhibition, calculating from curves obtained plotting inhibition percentage against essential oil concentration [35].

#### 4.3.2. Photochemiluminscence Assay

Photochemiluminescence (PCL) measures the antioxidant capacity of either lipophilic (ACL) or hydrophilic (ACW) pure compounds or complex mixtures. To measure the antioxidant activity of the essences, the (ACL) methodology was used as it was the most advisable to work with essential oils [35]. The PCL bioactivity of essential oil samples was compared to that of *T. vulgaris* essential oil, taken as positive control, and expressed as μmol of Trolox/mL [37].

#### 4.3.3. Qualitative Radical Scavenging Activity: HPTLC Bioautographic Assay

Bioautographic high performance thin layer chromatography (HPTLC) is an assay of antiradical activity that uses the DPPH and ABTS radicals to reveal the activity of the separate compounds or fractions in a complex mixture [34]. For the test, 30 μL of each essential oil was dissolved in 1 mL of methanol. 15 μL of these solutions were then applied directly to a Merck 60 HPTLC silica gel plate (Darmstadt, Germany), with F 254 fluorescence indicator, with a Camag LinomatV instrument (Muttenz, Switzerland). The mobile phase was *n*-hexane. In the developed plate, methanolic solutions of DPPH and ABTS were nebulized to determine the active fractions and to analyze their chemical composition with a subsequent analysis in GC-MS.

### 4.4. Antimicrobial Activity: Evaluation of the Minimum Inhibitory Concentration

The methodology known as disk diffusion was used, which is described in a number of investigations for essential oils [38,39,40]. The bacteria and yeasts used in the assays are listed below:

Gram positive bacteria: *Enterococcus faecalis* (ATCC 29212), *Listeria grayi* (ATCC 19120), *Micrococcus luteus* (ATCC 9622), *Staphylococcus aureus* (ATCC 29213), *Staphylococcus epidermidis (*DMS 20044), *Streptococcus mutans* (DMS 20523).

Gram negative bacteria: *Escherichia coli* (ATCC 4350), *Klebsiella oxytoca (*ATCC 29516), *Proteus vulgaris* (ATCC 6361), *Pseudomonas aeruginosa* (CBS 76039).

Yeasts: *Saccharomyces cerevisiae* (ATCC 2365), *Candida albicans (*ATCC 48274), *Malassezia furfur (*DSM 6170).

Mother cultures of each bacteria were set up 24 h before the assays in order to reach the stationary phase of growth. The tests were assessed by inoculating from the mother cultures Petri disks with proper sterile media with the aim of obtaining the microorganisms concentration 10^6^ CFU/mL. For bacteria, aliquots of dimethylsulfoxide (DMSO) were added to the essential oils in order to obtain a 0.05–500.0 mg/mL concentration range and then deposited on sterile paper disk (6 mm diameter, Becton Dickinson Italia S.p.A., Milan, Italy).

Bioactivity against the yeasts was also processed. Mother cultures were set up inoculating 100 mL Yeast Extract and Potato Dextrose (YEPD) liquid medium in 250 sterile flasks and for each mother culture at the stationary phase of growth, broth dilutions were made to obtain a strain concentration of 10^5^ CFU/mL to inoculate Petri dishes with agarized YEPD for bioassays. Then, 10 μL of DMSO-essential oil sample solutions were prepared in order to have an assay range 0.55–500 mg/mL, and then deposited on sterile paper disk (6 mm diameter, Difco). The Petri dishes were successively incubated at 30 °C in the dark and checked for evaluating the growth inhibition after 48 h, both for bacteria and yeasts streams: the lowest concentration of each essential oil showing a clear zone of inhibition was taken as the Minimum Inhibitory Concentration (MIC). Negative controls were set up with 10 μL of DMSO in the test solution, while positive ones were assessed with *T. vulgaris* essential oil [41]. Data reported for each assay are the average of three independent experiments.

#### Bioautographic Antimicrobial Activity of *H. coronarium* Essential Oil

The method foresees the development of a chromatographic plate, which is put in contact with the culture medium, the bacteria to be analyzed and the dye 2,3,5-triphenyltetrazolium cloride (TTC), which serves as a means of revealing the bacterial activity, where discoloration occurs at the time of cellular inactivity, as previously described [34,42]. A Merck HPTLC 60 silica gel plate was used to separate the compounds and toluene/ethylacetate/petroleum ether (93/7/20) mixture was used as mobile phase. The assay was performed with Gram positive bacteria *S. aureus* ATCC 6538 [34]. 10, 15 and 20 μL of a 30 μL/mL essential oil solution were applied on HPTLC.

### 4.5. Statistical Analysis

Relative standard deviations and statistical significance (Student’s *t* test; *p* < 0.05), one-way ANOVA and LSD post hoc Fisher’s honest significant difference test, were given, where appropriate, for all data collected. All computations were made using the statistical software STATISTICA 6.0 (StatSoft Italia srl, Vigonza, Italy).

## 5. Conclusions

Our research has focused on the chemical characterization and biological activity of five essential oils. In particular we studied for the first time the composition of two essential oils from leaves of *S. aspera* and *P. leticianum*, thus contributing to increasing the information about chemo-biodiversity in Ecuador. A second positive result arose from a performed antimicrobial evaluation, where *H. coronarium* has been proven as the most interesting essential oil: *P. augustum* and *P. leticianum* showed a high effect on *S. mutans* and all of them could be proposed as anticaries agenta, in agreement with their ancestral use [8]. Finally, it can be concluded that the valorization of these five species could in the near future become an alternative source of development funds for the communities that inhabit the Kutuku mountain range, as well as a starting point to investigate other species and evaluate various other types of applications in the pharmaceutical, cosmetic and food fields.

## Figures and Tables

**Figure 1 molecules-24-01637-f001:**
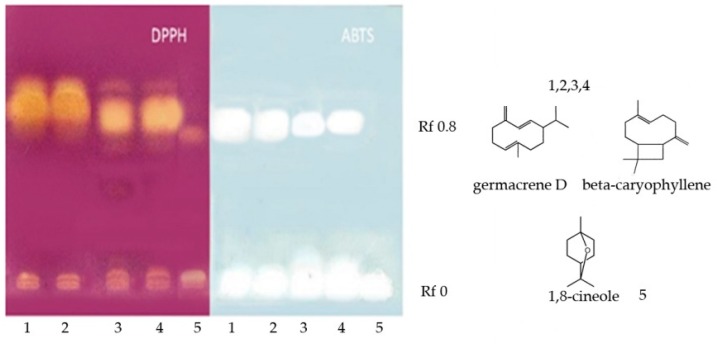
DPPH and ABTS bioautographic assay. **1**: *S. aspera*, **2**: *S. macrotepala*, **3**: *P. augustum*, **4**: *P. leticianum* and **5**: *H. coronarium*, 15 μL of essential oils (30 μL/mL).

**Figure 2 molecules-24-01637-f002:**
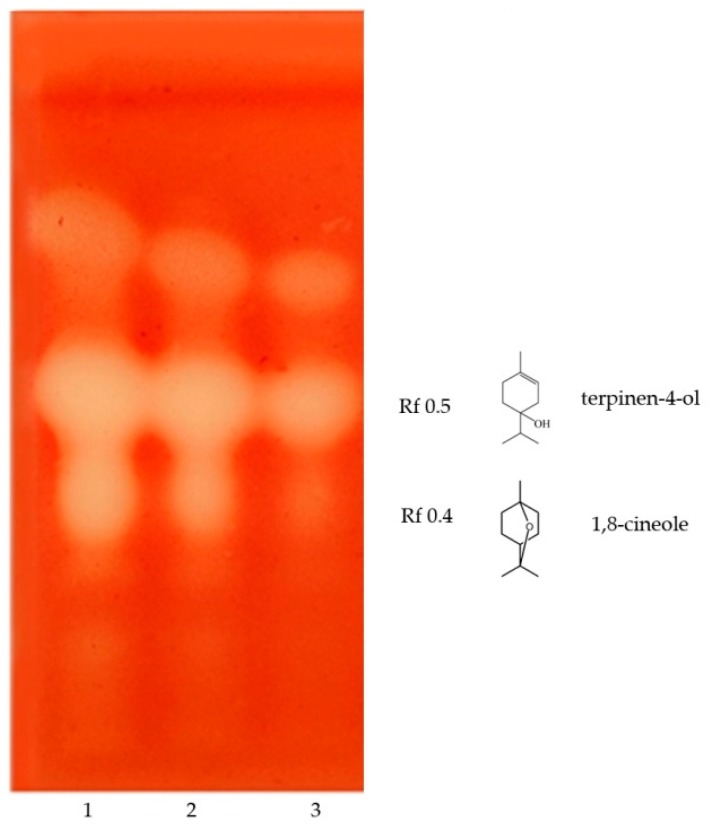
Bioautographic antibacterial assay of *H. coronarium* essential oil against *S.aureus*. 1:20 μL, 2:15 μL, 3:10 μL of *H.coronarium* essential oil (30 μL/mL).

**Table 1 molecules-24-01637-t001:** Chemical composition of essential oils.

Molecules	AI Lit ^a^	AI Exp ^b^	RA ^c^				
			*S. a*	*S. m*	*P. a*	*P. l*	*H. c* ^d^
α-pinene	932	930	7.0	1.8	0.4	1.5	10.0
camphene	946	946	0.3	0.2	-	-	0.7
sabinene ^e^	969	968	-	-	-	-	0.3
β-pinene ^e^	974	975	2.1	0.5	0.5	1.7	30.0
myrcene ^e^	990	991	0.1	0.1	-	0.1	0.5
α-terpinene	1017	1014	-	-	-	-	0.3
*p*-cymene ^e^	1020	1021	-	-	-	-	1.2
limonene ^e^	1024	1025	0.3	0.1	0.2	0.4	3.1
β-phellandrene	1025	1027	-	-	-	-	0.9
1,8-cineole ^e^	1026	1028	0.1	-	-	0.1	33.7
(*Z*)-β-ocimene ^e^	1032	1035	0.3	-	0.2	0.4	-
(*E*)-β-ocimene ^e^	1044	1050	-	-	1.8	3.5	-
γ-terpinene	1054	1052	-	-	-	-	1.2
*cis*-sabinene hydrate	1064	1067	-	-	-	-	0.1
terpinolene	1086	1081	-	-	-	-	0.3
linalool ^e^	1095	1102	-	-	-	-	0.5
perillene	1102	1112			-	0.7	-
*endo*-fenchol	1114	1115	-	-	-	-	0.1
*cis*-*p*-menth-2-en-1-ol	1118	1121	-	-	-	-	0.1
*trans*-pinocarveol	1135	1135	-	-	-	-	0.1
pinocarvone	1160	1157	-	-	-	-	0.1
borneol ^e^	1165	1166	-	-	-	-	1.1
terpinen-4-ol ^e^	1174	1175	-	-	-	-	2.4
α-terpineol ^e^	1186	1190	0.1	-	-	-	5.7
3,5-dimethoxytoluene	1269	1264	-	-	0.5	1.4	-
2-undecanone	1293	1293	-	0.3	-	-	-
δ-elemene	1335	1337	4.5	-	1.3	1.6	-
α-cubebene	1345	1351	1.7	1.8	0.3	0.3	-
α-terpinyl acetate	1346	1356	-	-	-	-	0.1
cyclosativene	1369	1370	0.2	0.1	-	-	-
α-ylangene	1373	1371	0.7	0.2	0.1	-	-
α-copaene	1374	1377	4.5	4.4	1.9	1.9	-
β-bourbonene	1387	1382	1.7	1.0	0.2	0.2	-
β-cubebene	1387	1388	0.3	1.9	0.2	0.4	-
*iso*-longilofolene	1389	1387	1.5	-	-	-	-
β-elemene^e^	1389	1391	2.3	1.5	5.8	5.1	-
β-longipinene	1400	1397	-	-	0.2	0.1	-
E-β-caryophyllene ^e^	1417	1411	3.3	3.4	27.1	21.8	0.4
β-copaene	1430	1425	1.0	1.0	0.7	0.5	-
β-gurjunene	1431	1427	0.1	-	-	-	-
γ-elemene	1434	1425	-	-	0.1	0.1	-
α-guaiene	1437	1433	1.0	0.7	-	-	-
aromandendrene	1439	1430	-	-	0.5	0.4	-
aristolene	1450	1439	0.5	-	-	-	-
*cis*-muurola-3,5-diene	1448	1446	0.4	0.1	0.2	0.3	-
*trans*-muurola-3,5-diene	1451	1454	-	0.4	-	-	-
α-humulene	1454	1451	1.2	0.8	3.1	2.9	0.1
*allo*-aromandendrene	1458	1453	0.2	0.5	0.6	0.6	-
dehydroaromadendrene	1460	1460	-	-	-	0.1	-
*cis*-cadina-1(6),4-diene	1461	1459	0.5	-	-	-	-
9-*epi*- β-caryophyllene	1464	1455	0.4	-	0.3	0.2	-
*cis*-muurola-4(14),5-diene	1465	1467	-	0.5	-	-	-
γ-gurjunene	1475	1471	0.7	-	0.7	0.5	-
γ-muurolene	1478	1475	2.2	0.5	1.9	1.3	-
germacrene D ^f^	1484	1480	23.3	42.1	11.2	9.0	-
β-selinene	1489	1485	0.9	0.5	1.8	1.5	0.1
drim-8(12)-ene	1491	1484	-	-	-	-	0.1
*trans*-muurola-4(14),5-diene	1493	1484	-	1.1	-	-	-
valencene	1496	1487	0.7	-	0.6	0.4	-
(*Z*,*E*)-α-farnesene	1491	1491			3.2	2.7	-
bicyclogermacrene	1500	1492	7.8	11.8	5.2	4.0	-
α-muurolene	1500	1495	1.1	1.2	1.3	0.8	-
β-himachalene	1500	1498	1.2	-	0.4	0.2	-
(*E,E*)-α-farnesene	1505	1505	-	0.2	5.6	5.1	-
germacrene A	1508	1501	1.1	-	2.4	2.6	-
γ-cadinene	1513	1508	4.3	1.4	1.4	0.7	-
cubebol	1514	1510	-	0.3	-	-	-
7-*epi*-γ-selinene	1522	1511	-	-	-	0.5	-
δ-cadinene	1522	1517	4.6	5.0	4.6	2.9	-
*cis*-calamenene	1528	1529	-	-	0.2	0.2	-
zonarene	1528	1530	-	0.2	-	-	-
*trans*-cadina-1(2),4 diene	1535	1531	0.3	0.4	0.3	0.2	-
α-cadinene	1537	1535	0.3	0.4	0.3	0.2	-
α-calacorene	1544	1540	0.3	0.1	-	-	-
germacrene B	1559	1557	1.3	1.7	1.2	1.2	-
E-nerolidol ^e^	1561	1563	-	-	0.5	1.7	-
spathulenol	1577	1577	1.2	0.8	0.6	0.8	-
caryophyllene oxide ^e^	1582	1581	0.2	0.1	0.8	3.8	0.2
globulol	1590	1585	-	0.5	-	-	-
viridiflorol	1592	1586	0.4	0.4	0.2	0.1	-
carotol	1594	1599	-	-	-	0.2	-
guaiol	1600	1597	0.4	0.4	0.1	0.2	-
β-oplopenone	1607	1609	0.5	0.1	-	-	-
humulene 1,2-epoxide	1608	1607	-	-	-	0.5	-
1,10-di-*epi*-cubenol	1618	1617	0.1	0.2	-	0.1	-
10-*epi*-γ-eudesmol	1622	1617	-	-	-	0.1	-
1-*epi*-cubenol	1627	1630	0.4	0.6	0.4	0.5	-
*epi*-α-cadinol	1638	1646	0.7	0.7	0.5	0.4	-
*epi*-α-muurolol	1640	1648	0.5	0.7	0.7	0.7	-
α-muurolol	1644	1651	0.8	0.5	0.4	0.5	-
α-cadinol	1652	1660	1.2	1.5	0.8	0.7	-
selin-11-en-4-α-ol	1658	1660	-	-	0.6	0.7	-
intermedeol	1665	1668	-	-	0.3	0.5	-
khusinol	1679	1689	0.4	-	-	-	-
eudesma-4(15),7-dien-1-β-ol	1687	1696	0.2	-	-	-	-
cyclocolorenone	1759	1761	-	-	0.1	0.1	-
Total identified (%)			93.1	94.7	94.5	91.9	93.4

^a^ Literature arithmetic index by Adams [22], ^b^ Experimental arithmetic index, ^c^ Relative area (%), ^d^
*S. a*: *S. aspera*, *S. m*: *S. macrotepala*, *P. a*: *P. augustum*, *P. l*: *P. leticianum*: *H. c*: *H. coronarium*, ^e^ co-injected pure compounds, ^f 1^H- and ^13^C- NMR spectra are reported as Appendix A [23]. All %area had a standard deviation < 5.0%.

**Table 2 molecules-24-01637-t002:** Free radical scavenging activity of the essential oils evaluated by DPPH and ABTS spectrophotometric methods.

Essential Oils and Pure Molecules	IC_50_ mg/mL	
	DPPH	ABTS
*S. aspera*	20.70 ± 0.80	1.12 ± 0.04
*S. macrotepala*	29.37 ± 1.15	0.80 ± 0.03
*P. augustum*	6.17 ± 0.33	2.16 ± 0.20
*P. leticianum*	4.26 ± 0.11	2.65 ± 0.25
*H. coronarium*	9.04 ± 0.55	2.87 ± 0.17
*T. vulgaris*	0.71 ± 0.02	0.055 ± 0.001
*E*-β-caryophyllene	80.1 ± 1.40	15.1 ± 1.16
β-pinene	149.8 ± 5.66	142.0 ± 9.07
1,8-cineole	440.8 ± 10.18	174.1 ± 7.44
germacrene D	2.1 ± 0.02	1.19 ± 0.02

**Table 3 molecules-24-01637-t003:** Results of the antioxidant activity of essential oils by the ACL methodology.

Essential Oils	μmol of Trolox/mL (*p* ≤ 0.05)
*S. aspera*	4.72 ± 0.08
*S. macrotepala*	5.43 ± 0.15
*P. augustum*	1.07 ± 0.03
*P. leticianum*	1.35 ± 0.04
*H. coronarium*	9.04 ± 0.05
*T. vulgaris*	283.33 ± 8.57

**Table 4 molecules-24-01637-t004:** Antimicrobial activity expressed with minimum inhibitory concentration (MIC mg/mL).

Microorganism		*S. aspera* MIC (mg/mL)	*S. macrotepala* MIC (mg/mL)	*P. augustum* MIC (mg/mL)	*P. leticianum* MIC (mg/mL)	*H coronarium* MIC (mg/mL)	*T. vulgaris* MIC (mg/mL)
Gram + bacteria	EF	9.3	9.0	9.1	9.1	9.0	1.8
	LIST	9.3	9.3	18.2	18.1	0.45	0.9
	MLU	4.6	18.6	18.2	91.1	9.0	1.8
	SAU	46.0	46.5	91.0	18.1	9.0	1.8
	SE	18.6	18.6	18.2	18.1	4.5	0.9
	SMU	1.9	0.9	0.18	-0.18	0.18	0.18
Gram − bacteria	EC	464	465	454	453	89.5	4.6
	KOX	18.6	46.5	45.4	45.3	0.9	0.9
	PVU	18.6	46.5	45.4	18.1	9.0	0.9
	PA	464	93.0	91.0	9.,6	89.5	9.2
Yeasts	SC	92.9	465.0	18.2	18.1	89.5	1.8
	CAND	46.0	93.0	91.0	45.3	17.9	1.8
	MF	18.6	46.5	1.8	18.1	4.5	0.18

Note: EF = Enterococcus faecalis; LIST = Listeria grayi; MLU = Micrococcus luteus; SAU = Staphylococcus aureus; SE = Staphylococcus epidermidis; SMU = Streptococcus mutans; EC = Escherichia coli; KOX = Klebsiella oxytoca; PVU = Proteus vulgaris; PA = Pseudomonas aeruginosa; SC = Saccharomyces cerevisiae; CAND = Candida albicans; MF = Malassezia furfur. All MIC values had a standard deviation < 10.0%.

**Table 5 molecules-24-01637-t005:** Collection sites for plant species.

Species	Site Collection	Geographical Coordinates
*S. aspera*	San Luis del Upano parish, Morona Santiago province.	Latitude: S 2°28′43″ Length: W 78°8′59″ Altitude: 820 msm
*S. macrotepala*	Shakaim Biological Station, Chiguaza parish, Morona Santiago province.	Latitude: S 02°03′52.2″, Length: W 77°52′32.5″ Altitude: 1200 msm
*P. augustum*	Shakaim Biological Station, Chiguaza parish, Morona Santiago province.	Latitude: S 02°03′52.2″, Length: W 77°52′32.5″ Altitude: 1200 msm
*P. leticianum*	Shakaim Biological Station, Chiguaza parish, Morona Santiago province.	Latitude: S 02°03′52.2″, Length: W 77°52′32.5″ Altitude: 1200 msm
*H. coronarium*	Macas, Morona Santiago province	Latitude: S 2°10′ Length: W 78°0′ Altitude: 1080 msm

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
