# Peer review of "Chemical Composition and Biological Activity of Five Essential Oils from the Ecuadorian Amazon Rain Forest"

_molecules, 2019, doi:10.3390/molecules24081637_

Round 1

Reviewer 1 Report

Manuscript presents research on composition and biological activity of five essential oils(EOs) two of them were not known previously. Experiments were properly schemed and well done. I especially appreciate the use of thyme oil as a reference in assessment of biological activities. Manuscript is in general well organized however, English needs correction, especially in Discussion. Some sentences are understandable.  Results  are worth publishing. However, there are some inaccuracies that need improving.

I present my remarks below and mark some of them in the manuscript.

Materials and Methods

Give please the amount of plant material subjected to hydrodistillation.

I would like Authors to be aware that the true percentages of EOs components can be obtained by GC with FID detector but not by MS detector. Chapter 2.3.1. needs correction  of concentration of DPPH and ABTS as well as of formula.

In chapter 2.4. antimicrobial assay should be described more precisely, especially the way of MIC assessment should be added. Bioautography in antimicrobial activity should be described.

Please do not use the term "extraction" to hydrodistillation process (ls 14, 64, 94).

Table 2

It is hardly possible to distillate cis-retinal because of its low volatility.  What is more the experimental retention index of compound identified as cis-retinal (2130) is much lower than literature one (2466). Hence, this must be misidentification.

The percentages of EOs constituents should be rounded to 0.1%  (one decimal place).

For retention indices abbreviation RI is used.

There are many falsely written constituents names, I give proper ones:  camphene, myrcene, 1,8-cineole, (Z)-β-ocimene, (E)- β-ocimene, cis-p-menth-2-en-1-ol, 3,5-dimethoxytoluene, β -bourbonene, isolongifolene, drim-8,(12)-ene, trans-muurola-4(14),5-diene, (Z,E)-α-farnesene, (E,E)-α-farnesene, trans-cadina-1(2),4-diene, α-calacorene, (E)-nerolidol, caryophyllene oxide, humulene epoxide, selin-11-en-4α-ol, eudesma...dien-1β-ol, total identified. Prefixes such as cis, trans, allo, epi should be written in Italic; the same apply to Table 3.

Insert lit. RI for (Z,E)-α-farnesene, and exp. RI for drimene.

In legend to Table 2: literature RI by Adams not theoretical ones are given.

In chapter 2.3.3. the amount of EO spread on TLC plate is reported properly,  in Figure 1 instead of amount concentration is reported and this should be changed.  The same apply to Figure 2 - it is important haw much EO was spread.

Antileishmaniasis properties (ls. 61, 66) doesn't means the same as leishmanicidal (Ref. 6) or antileishmanial (ref. 9), so please change this.

English in titles of Figures 1 and 2, chapter 2.5. should be corrected additionally to corrections in all text.

Results

In Table 3 four EO constituents are listed. Nothing about them is reported in Materials and Methods. Did you really use standards?

The interpretation of  EO TLC in hexane is false (Figure 1). In these conditions all hydrocarbos, both mono- and sesquiterpene have Rf about 0.8 (among them β-caryophyllene) while oxygenated terpenes have low Rf (0-0.1).  Antioxidant activity could be attributed to these two group of compounds unless you use pure reference compounds.

Author Response

-     Manuscript presents research on composition and biological activity of five essential oils(EOs) two of them were not known previously. Experiments were properly schemed and well done. I especially appreciate the use of thyme oil as a reference in assessment of biological activities. Manuscript is in general well organized however, English needs correction, especially in Discussion. Some sentences are understandable.  Results  are worth publishing. However, there are some inaccuracies that need improving.

I present my remarks below and mark some of them in the manuscript.

We agree with your recommendations: we have revised the manuscript following all your suggestions. We have rewritten some sentences and better discussed the results.

Materials and Methods

-     Give please the amount of plant material subjected to hydrodistillation.

-     I would like Authors to be aware that the true percentages of EOs components can be obtained by GC with FID detector but not by MS detector. Chapter 2.3.1. needs correction  of concentration of DPPH and ABTS as well as of formula.

According to the reviewer there are many mistakes in the paper. Quantification was determined by GC-FID, as described in our previous papers (e.g., Scalvenzi, L.;., Grandini, A., Spagnoletti, A.;, Tacchini, M.;, Neill, D.;, Ballesteros, J.L.;, Sacchetti, G.; Guerrini, A. Myrcia splendens (Sw.) DC. (syn. M. fallax (Rich.) DC.) (Myrtaceace) essential oil from Amazonian Ecuador: a chemical characterization and bioactive profile. Molecules 2017, 22(7), 1163). We implemented “material and methods” section with these informations. We have changed the test and clarified in Table 2.

Chapter 2.3.1. has been corrected

-     In chapter 2.4. antimicrobial assay should be described more precisely, especially the way of MIC assessment should be added. Bioautography in antimicrobial activity should be described.

We have better described methods in Chapter 2.4

-     Please do not use the term "extraction" to hydrodistillation process (ls 14, 64, 94).

We have changed extraction with isolation

-     Table 2

It is hardly possible to distillate cis-retinal because of its low volatility.  What is more the experimental retention index of compound identified as cis-retinal (2130) is much lower than literature one (2466). Hence, this must be misidentification.

We agree: we have deleted this possible identification and have also changed Figure 2

-     The percentages of EOs constituents should be rounded to 0.1%  (one decimal place).

We agree: we have changed all  % areas with a one decimal place.

-     For retention indices abbreviation RI is used.

We have used AI as proposed by Adams, 2007.

-     There are many falsely written constituents names, I give proper ones:  camphene, myrcene, 1,8-cineole, (Z)-β-ocimene, (E)- β-ocimene, cis-p-menth-2-en-1-ol, 3,5-dimethoxytoluene, β -bourbonene, isolongifolene, drim-8,(12)-ene, trans-muurola-4(14),5-diene, (Z,E)-α-farnesene, (E,E)-α-farnesene, trans-cadina-1(2),4-diene, α-calacorene, (E)-nerolidol, caryophyllene oxide, humulene epoxide, selin-11-en-4α-ol, eudesma...dien-1β-ol, total identified. Prefixes such as cistransalloepi should be written in Italic; the same apply to Table 3.

Insert lit. RI for (Z,E)-α-farnesene, and exp. RI for drimene.

In legend to Table 2: literature RI by Adams not theoretical ones are given.

We have changed all names and definitions according your suggestions and inserted RI (AI) values.

-       In chapter 2.3.3. the amount of EO spread on TLC plate is reported properly,  in Figure 1 instead of amount concentration is reported and this should be changed.  The same apply to Figure 2 - it is important haw much EO was spread.

We have reported the correct amount in “material and methods”, as well as in Figure 1 and Figure 2

-       Antileishmaniasis properties (ls. 61, 66) doesn't means the same as leishmanicidal (Ref. 6) or antileishmanial (ref. 9), so please change this.

We have changed according literature.

-       English in titles of Figures 1 and 2, chapter 2.5. should be corrected additionally to corrections in all text

We have changed titles and text.

Results

-       In Table 3 four EO constituents are listed. Nothing about them is reported in Materials and Methods. Did you really use standards?

We have used the four constituents, so we have properly changed “Materials and Methods”

-       The interpretation of  EO TLC in hexane is false (Figure 1). In these conditions all hydrocarbos, both mono- and sesquiterpene have Rf about 0.8 (among them β-caryophyllene) while oxygenated terpenes have low Rf (0-0.1).  Antioxidant activity could be attributed to these two group of compounds unless you use pure reference compounds.

We agree: we have checked our data and are congruent with your observations. We have changed results and Figure 1.

Reviewer 2 Report

In the article entitled "Chemical Composition and Biological Activity of Six Essential Oils from Ecuadorian Amazon Rain Forest ", the authors established the composition, the antioxidant properties and biological activities. Different magnitudes of potency of biological activities were detected according to the analyzed essential oils, to the performed tests, to the main components of essential oils or to the synergistic effects of different volatile constituents.

There are few studies on essential oil composition and their biological activity. However, the results are mostly in line with previous reports, and no contains new sing for readers.

In my opinion, this manuscript do not meet the requirements of the Molecules.

The identification of the volatiles does not meet the standards of a chemistry journal, because it was not performed according to state of the art methods in volatile analysis (apolar and polar columns, literature index for comparison). GC (RI) is not used in this study and quantification with GC/MS is not adequate.

Many mistakes have no title in a scientific paper, and finally there are too many typing errors, which renders the impression of a great carelessness in proof reading before submission of the manuscript:

Camphene instead of canpheno

myrcene instead of mirceno

 (Z)-b-ocimene instead of cis- ocimene

 (E)-b-ocimene instead of trans-ocimene

 cis-p-menth-2-en-1-ol instead of cis-p-ment-2-en1-ol

terpinen-4-ol instead of 4-terpineol

 3,5-dimethoxy toluene instead of 3,5-dimetoxi toluene

 β-bourbonene instead of β- bourbunene

 bicyclogermacrene instead of byciclogermacrene

β-himachalene instead of β- himanchalene

humulene 1,2-epoxide instead of humulene 1,2-epoxidoe

selin-11-en-4-a-ol instead of selin-11-en-4-a-ol

identified instead of identificado

Table 3

 1,8 cineole instead of 1,8 cineol

β-pinene instead of β-pineno

 (E)-b-caryophyllene instead of β-cariofileno

 Germacreno D instead of germacreno D

Author Response

-     In the article entitled "Chemical Composition and Biological Activity of Six Essential Oils from Ecuadorian Amazon Rain Forest ", the authors established the composition, the antioxidant properties and biological activities. Different magnitudes of potency of biological activities were detected according to the analyzed essential oils, to the performed tests, to the main components of essential oils or to the synergistic effects of different volatile constituents.

There are few studies on essential oil composition and their biological activity. However, the results are mostly in line with previous reports, and no contains new sing for readers.

In my opinion, this manuscript do not meet the requirements of the Molecules.

We have implemented all the sections of paper to improve its quality

-       The identification of the volatiles does not meet the standards of a chemistry journal, because it was not performed according to state of the art methods in volatile analysis (apolar and polar columns, literature index for comparison). GC (RI) is not used in this study and quantification with GC/MS is not adequate.

According to the reviewer there are many mistakes in the paper. We have used AI (arithmetic Index) as proposed in Adams, 2007 (see reference in the paper). We have changed the text in the paper and clarified in Table 2. Quantification was determined by GC-FID, as described in our previous papers (e.g., Scalvenzi, L.;., Grandini, A., Spagnoletti, A.;, Tacchini, M.;, Neill, D.;, Ballesteros, J.L.;, Sacchetti, G.; Guerrini, A. Myrcia splendens (Sw.) DC. (syn. M. fallax (Rich.) DC.) (Myrtaceace) essential oil from Amazonian Ecuador: a chemical characterization and bioactive profile. Molecules 2017, 22(7), 1163). We have implemented “material and methods” section with these informations.

Many authors reported literature and experimental arithmetic index instead of apolar and polar retention index (e.g., Sandra Layse F. Sarrazin, Leomara Andrade da Silva, Ana Paula F. de Assunção, Ricardo B. Oliveira, Victor Y. P. Calao, Rodrigo da Silva, Elena E. Stashenko, José Guilherme S. Maia, Rosa Helena V. Mourão Antimicrobial and Seasonal Evaluation of the Carvacrol-Chemotype Oil from Lippia origanoides Kunth. Molecules. 2015, 20(2), 1860–1871)

-       Many mistakes have no title in a scientific paper, and finally there are too many typing errors, which renders the impression of a great carelessness in proof reading before submission of the manuscript:

Camphene instead of canpheno

myrcene instead of mirceno

 (Z)-b-ocimene instead of cis- ocimene

 (E)-b-ocimene instead of trans-ocimene

 cis-p-menth-2-en-1-ol instead of cis-p-ment-2-en1-ol

terpinen-4-ol instead of 4-terpineol

 3,5-dimethoxy toluene instead of 3,5-dimetoxi toluene

 β-bourbonene instead of β- bourbunene

 bicyclogermacrene instead of byciclogermacrene

β-himachalene instead of β- himanchalene

humulene 1,2-epoxide instead of humulene 1,2-epoxidoe

selin-11-en-4-a-ol instead of selin-11-en-4-a-ol

identified instead of identificado

We have corrected all the mystakes

-       Table 3

 1,8 cineole instead of 1,8 cineol

β-pinene instead of β-pineno

 (E)-b-caryophyllene instead of β-cariofileno

 Germacreno D instead of germacreno D

We have corrected all the mystakes

Reviewer 3 Report

Currently, there is a return to natural medicine and that's why essential oils are so popular. Therefore, it is very important to know their chemical composition.

This manuscript describing differences in chemical composition and activity of six essential oils.

I noted the following:

An adequate statistical analysis of the results is required to significantly improve the manuscript.

The University of Ferrara was given in the list of affiliations. However, the affiliations of the authors are: Universidad Politécnica Salesiana and Shakaim Biological Station.

Is the name “limoncello” used for both Siparuna aspera and Siparuna macrotepala?

Scientific names should be in italics – abstract and references

A lot of errors during text conversion:

-          too large spaces between words: line 60, 65, 98, 111, 120,258-259, 334-335, table 2

-          different font size: line 38, 108-109, 148-155

-          subscripts: line 119, 124, 194-195, 181, 182, 332

Errors in compound names:

-          instead “1.8 cineole” should be “1,8-cineole” (line 33, table 2)

-          instead “1.8 cineol” should be “1,8-cineole” (line 196, 213)

-          instead “1-8 cineole” should be “1,8-cineole” (line 232)

-          instead “1,8 cineole” should be “1,8-cineole” (line 239)

-          instead “1.8-cineole” should be “1,8-cineole” (line 344)

-          instead “canpheno” should be “camphene” (table 2)

-          instead “mirceno” should be “myrcene” (table 2)

-          instead “3,5-dimetoxi toluene” should be “3,5-dimethoxytoluene” (table 2)

The names of compounds in the table 2 should be written with a lowercase letter.

Instead “total identificado” in table 2 should be “total identified”.

Instead “polysiloxane ) of 30m length” in line 100 should be “polysiloxane of 30m length”.

Instead “35-400 Da ,trap” in line 106 should be “35-400 Da, trap”.

Instead “temperature:260” in line 106 should be “temperature: 260”.

The reference 24 is given as the superscript in line 110.

Author Response

Currently, there is a return to natural medicine and that's why essential oils are so popular. Therefore, it is very important to know their chemical composition.

This manuscript describing differences in chemical composition and activity of six essential oils.

I noted the following:

-       An adequate statistical analysis of the results is required to significantly improve the manuscript.

We have added statistical analysis in the text

-       The University of Ferrara was given in the list of affiliations. However, the affiliations of the authors are: Universidad Politécnica Salesiana and Shakaim Biological Station.

-       Is the name “limoncello” used for both Siparuna aspera and Siparuna macrotepala?

-       Scientific names should be in italics – abstract and references

We have checked and corrected

-  A lot of errors during text conversion:

-          too large spaces between words: line 60, 65, 98, 111, 120,258-259, 334-335, table 2

-          different font size: line 38, 108-109, 148-155

-          subscripts: line 119, 124, 194-195, 181, 182, 332

We have checked and corrected

- Errors in compound names:

-          instead “1.8 cineole” should be “1,8-cineole” (line 33, table 2)

-          instead “1.8 cineol” should be “1,8-cineole” (line 196, 213)

-          instead “1-8 cineole” should be “1,8-cineole” (line 232)

-          instead “1,8 cineole” should be “1,8-cineole” (line 239)

-          instead “1.8-cineole” should be “1,8-cineole” (line 344)

-          instead “canpheno” should be “camphene” (table 2)

-          instead “mirceno” should be “myrcene” (table 2)

-          instead “3,5-dimetoxi toluene” should be “3,5-dimethoxytoluene” (table 2)

We have checked and corrected

-       The names of compounds in the table 2 should be written with a lowercase letter.

Instead “total identificado” in table 2 should be “total identified”.

Instead “polysiloxane ) of 30m length” in line 100 should be “polysiloxane of 30m length”.

Instead “35-400 Da ,trap” in line 106 should be “35-400 Da, trap”.

Instead “temperature:260” in line 106 should be “temperature: 260”.

The reference 24 is given as the superscript in line 110.

We have checked and corrected

Round 2

Reviewer 2 Report

The manuscript has been modified according to all suggestions and requirements of referees. However, the identification of the volatiles does not meet the standards of a chemistry journal. Two GC columns (apolar and polar) must be used for essential oil analysis. Identification of oxygenated sesquiterpenes particularly epimers such as epi-α-muurolol and α-muurolol with one column is not usual.

The component identification must use at least two methods to validate identity, most commonly, mass spectrometry and gas chromatography (e.g. MS spectra and retention indices (RIs)). The methodology of analysis developed herein creates doubts. For instance, cis and trans-calamenene exhibited the same RI and the same mass spectra. Then, the identification of trans-calamenene is not possible using the described parameters.

The quality of the submission concerning the biological activities is high however the analytical methods for chemical composition are border line and must be improved.

 The quality of molécules in figures1 and 2 is bad.

Author Response

Dear Editor,

We have reorganized the manuscript according the template suggested by the Journal. We checked English. We replied to point-by-point to the reviewer’s comments, below.

The manuscript has been modified according to all suggestions and requirements of referees. However, the identification of the volatiles does not meet the standards of a chemistry journal. Two GC columns (apolar and polar) must be used for essential oil analysis. Identification of oxygenated sesquiterpenes particularly epimers such as epi-α-muurolol and α-muurolol with one column is not usual.

We thank the reviewer for his comments. However, “Molecules” Journal in the section “Correct Identification of Natural Products” suggests the following Instruction for Authors: “The identification of known compounds in extracts should be safely supported with chromatographic and/or spectroscopic data (e.g. LC/GC retention times and/or UV, mass spectra), as well as comparison with data of authentic samples or previously published values. When previously reported compounds are isolated and used in biological activity assays, the 1H NMR spectrum should be given in supplementary data as a proof of purity” (https://www.mdpi.com/journal/molecules/instructions#natural_products)

Consequently, no mention is made about the use of two columns for identification of known compounds. In fact, in recent published papers, reporting the identification of compounds in essential oils, several authors performed the chemical characterization with a single column: Molecules 2015, 20, 7034-7047; doi:10.3390/molecules20047034, Molecules 2019, 24, 1394; doi:10.3390/molecules24071394; Molecules 2019, 24, 1206; doi:10.3390/molecules24071206; Molecules 2019, 24, 1062; doi:10.3390/molecules24061062

We also published in this Journal using the same methodology proposed in our present article: Molecules 2017, 22, 1163; doi:10.3390/molecules22071163.

According guidelines, we used the co-injection of pure standards to define the main components, as indicated in Materials and Methods: We improved table 1 adding the indication of which pure compounds were injected.

We instead added the 1H e 13C NMR spectrum of germacrene D, isolated and tested for its biological properties, as supplementary material. Data were in according to those previously published: T. RøstelienA.-K. Borg-KarlsonJ. FäldtU. JacobssonH. Mustaparta (2000) The Plant Sesquiterpene Germacrene D Specifically Activates a Major Type of Antennal Receptor Neuron of the Tobacco Budworm Moth Heliothis virescensChemical Senses, 25(2), 141–148,

Regarding the identification of oxygenated sesquiterpenes particularly epimers such as epi-α-muurolol and α-muurolol with one column, we agree with the referee, but the epimers identified in our described essential oils are minor constituents and their unambiguous identification could even be difficult with a second column. Only a NMR experiment could certainty clarify the structure: we had small quantities of essential oils and so we had no possibility to perform a separation of minor compounds for a NMR analysis.

The component identification must use at least two methods to validate identity, most commonly, mass spectrometry and gas chromatography (e.g. MS spectra and retention indices (RIs)). The methodology of analysis developed herein creates doubts. For instance, cis and trans-calamenene exhibited the same RI and the same mass spectra. Then, the identification of trans-calamenene is not possible using the described parameters.

Sorry, the identification of trans-calamenene is a mistake, a typing errorWe change AI and the name as cis-calamenene

The quality of the submission concerning the biological activities is high however the analytical methods for chemical composition are border line and must be improved.

We have improved the table 1, adding the indication that, as in our other previously works, we used also the co-injection of pure standards to define the main components (J. L. Ballesteros, M. Tacchini, A. Spagnoletti, A. Grandini, G. Paganetto, L. M. Neri, A. Marengo, L. Angiolella, A. Guerrini, G. Sacchetti (2019) Rediscovering Medicinal Amazonian Aromatic Plants: Piper carpunya (Piperaceae) Essential Oil as Paradigmatic Study, Evidence-Based Complementary and Alternative Medicine, ID 6194640

 The quality of molécules in figures1 and 2 is bad.

We have increased the quality
